ecology, plant science, environmental science

British Isles, citizen science, climate change, ecosystem service, plant phenology, woodland trust

**Author for correspondence:**
Ulf Büntgen
e-mail: ulf.buentgen@geog.cam.ac.uk

# Plants in the UK flower a month earlier under recent warming

Ulf Büntgen[1,3,4,5], Alma Piermattei[1], Paul J. Krusic[1,6], Jan Esper[4,7], Tim Sparks[2,8] and Alan Crivellaro[1,9]

[1]Department of Geography, and [2]Museum of Zoology University of Cambridge, Cambridge CB2 3EN, UK
[3]Swiss Federal Research Institute (WSL), 8903 Birmensdorf, Switzerland
[4]Global Change Research Institute of the Czech Academy of Sciences (CzechGlobe), 60300 Brno, Czech Republic
[5]Department of Geography, Faculty of Science, Masaryk University, 61300 Brno, Czech Republic
[6]Department of Physical Geography, Stockholm University, 10691 Stockholm, Sweden
[7]Department of Geography, Johannes Gutenberg University, 55099 Mainz, Germany
[8]Department of Zoology, Poznań University of Life Sciences, 60-625 Poznań, Poland
[9]Forest Biometrics Laboratory, Faculty of Forestry, 'Stefan cel Mare' University of Suceava. Str. Universitatii 13, Suceava 720229, Romania

UB, 0000-0002-3821-0818

Global temperatures are rising at an unprecedented rate, but environmental responses are often difficult to recognize and quantify. Long-term observations of plant phenology, the annually recurring sequence of plant developmental stages, can provide sensitive measures of climate change and important information for ecosystem services. Here, we present 419 354 recordings of the first flowering date from 406 plant species in the UK between 1753 and 2019 CE. Community-wide first flowering advanced by almost one month on average when comparing all observations before and after 1986 ($p < 0.0001$). The mean first flowering time is 6 days earlier in southern than northern sites, 5 days earlier under urban than rural settings, and 1 day earlier at lower than higher elevations. Compared to trees and shrubs, the largest lifeform-specific phenological shift of 32 days is found in herbs, which are generally characterized by fast turnover rates and potentially high levels of genetic adaptation. Correlated with January–April maximum temperatures at −0.81 from 1952–2019 ($p < 0.0001$), the observed trends (5.4 days per decade) and extremes (66 days between the earliest and latest annual mean) in the UK's first flowering dataset can affect the functioning and productivity of ecosystems and agriculture.

## 1. Introduction

The world's longest running and best-documented meteorological record, the Central England Temperature series [1], places the recent anthropogenic warming trend [2] as unprecedented in the context of natural climate variability of the past three and a half centuries (electronic supplementary material, figure S1). While the impact of rising mean temperatures and associated climatic extremes can be manifest in distinct environmental responses and societal consequences [3,4], the effects of long-term climate change on the functioning and productivity of biological, ecological and agricultural systems are often subtle, thus difficult to recognize and quantify [5]. This is particularly true in ecological research where the climatological concept of 'detection and attribution' has only recently been applied [6,7], as well as for farmers, policymakers and the wider public since statistical significance alone often remains an abstract dimension.

Changes in the timing and intensity of annually recurring patterns in biological systems, including the growth and development of plants and the behaviour of animals [8], are closely related to high-frequency climate variability [9]. Though global warming has been shown to alter the occurrence of

important developmental and behavioural events in birds, insects, amphibians and plants [8], most studies have involved single or small sets of species at local or regional scales. The history of observing and recording phenological events has been useful for informing environmental scientists, conservation agencies and policymakers about possible extinction risks and the loss of ecosystem services [10], for adapting agricultural techniques to prolonged growing seasons [7], and for improving respiratory allergy prevention and management [11,12]. Our understanding of the direct and indirect biophysical feedbacks of plant phenological changes on land surface-atmosphere exchanges is, however, the hardest to grasp and simultaneously among the most important to consider [8]. Though ground-based plant phenological observations and associated laboratory experiments can be supplemented with similar evidence from satellite remote-sensing over the past few decades [13], the continuous update, re-assessment and cross-comparison of species-specific, long plant phenological datasets describe a central task for the emerging, international and interdisciplinary arena of global change research. The scientific value of hundreds of thousands of observations of first flowering dates from a wide range of plant species is related to the high temperature sensitivity of this phenological event, the large sample size of citizen science datasets, and the spatio-temporal precision of the observed intra- and interannual changes (see [8] for an extensive review). However, such studies spanning longer time scales and larger biogeographic regions, operating at a community level, and considering a multitude of climatic variables, are particularly rare.

Here, we present the analysis of 419 354 observations of the first flowering date (FFD) from 406 plant species in the UK between 1753 and 2019 CE (figure 1 and table 1). This version of the UK's Phenology Network [10] contains daily resolved and spatially explicit observations from trees (138 382), shrubs (40 063), herbs (228 093) and climbers (12 816), which were collected between the Channel Islands in the south (49°10′ N), Shetland in the north (60°50′N), Northern Ireland in the west (8°10′ W), and Suffolk in the east (1°45′ E). After data homogenization and timeseries investigation of the FFDs, we use monthly resolved climate parameters and synoptic weather indices to quantify the direct and indirect drivers of plant phenological changes over space and time in the UK between 1753 and 2019 CE. We then discuss the implications of trends and extremes in FFDs for the functioning and productivity of biological, ecological and agricultural systems.

## 2. Methods

We obtained a total of 435 029 spatio-temporally explicit recordings of FFDs from 406 plant species between 1753 and 2019 at 6279 disjunct locations in the UK (figure 1a for the distribution of observation points), including the Crown dependencies of Jersey, Guernsey and the Isle of Man, hereafter called UK purely for brevity. All data were provided by the UK Phenology Network in the Nature's Calendar of the Woodland Trust (electronic supplementary material). We removed or corrected 15 511 phenological observations from the initial dataset because they were either duplicates (2923 observations), or multiple observations of species/year/location combinations where we retained only the earliest date (12 588 observations). The phenological information from another 35 observations that had geographical coordinates in the sea was deleted. For a more

general description of the phenological dataset used in this study and extracted from Nature's Calendar in July 2020, we refer to Amano et al. [10] and Collinson & Sparks [14].

Height estimates for adult plants of each species were used for lifeform classification: trees (greater than 4 m), shrubs (20 cm to 4 m), herbs (less than 20 cm) and climbers (no self-supporting plants). The species-specific range of the FFDs was defined by the first and last observation dates, which may start as early as the autumn of the previous year and end as late as December of the current year. The FFDs are therefore expressed on a scale from −100 DOY to 365 DOY, where 1 DOY corresponds to 1 January. After converting species FFDs to DOY, five tree species exhibit negative DOYs: Corylus avellana (210 observations), Prunus spinosa (six observations), Salix caprea (three observations) and Alnus glutinosa and Crataegus monogyna (one observation each); one shrub species exhibits negative DOYs: Daphne laureola (three observations); and 37 herb species exhibit negative DOYs: Helleborus foetidus (679 observations), Galanthus nivalis (227 observations), Eranthis hyemalis (45 observations), Primula vulgaris (83 observations), Ficaria verna (69 observations), Mercurialis perennis (12 observations), Tussilago farfara (nine observations), Petasites hybridus (seven observations), as well as another herb 29 species (with only one or two observations).

The 332 107 phenological observations less than 53.5°N were classified as 'southern' (south) and the 87 247 > 53.5°N were considered as 'northern' (north) based on climatological differences over the British Isles. As a rule of thumb, the northwest is characterized by mild winters and cool summers, the northeast is characterized by cold winters and cool summers, the southwest is characterized by mild winters and warm summers, and the southeast is characterized by cold winters and warm summers. While the west has a more maritime climate during winter, the east is often affected by cold airflow from the European continent. The latitudinal threshold around Manchester is further informed by floristic regions and the distribution of species in our phenological dataset. We also divided all FFDs into 251 315 and 168 039 observations from below and above 82 m sea level, respectively, which is the elevational mean of the total dataset. Landcover units from the CORINE version 2018 (https://land.copernicus.eu/pan-european/corine-land-cover) were assigned to each observation based on their geographical coordinates using the R package raster [15] and sp [16]. Each of the 44 CORINE landcover classifications was grouped into five macro categories: artificial surfaces (239 457 FFDs), agricultural areas (154 585 FFDs), forest and semi-natural areas (18 979 FFDs), water bodies (4621 FFDs) and wetlands (1712 FFDs). We further differentiated the phenological observations into either urban (232 291) or rural (179 897) sites, excluding a total of 7166 data points from either indistinguishable or hybrid site type classifications. Site elevation was extracted from WorldClim version 2.1. (https://www.worldclim.org/data/worldclim21.html), which is available at 30 s spatial resolution (i.e. at approximately 1 km²). Although the mean, median and mode were separately considered in each analysis step, only the mean is shown because in nearly all instances their values are similar. The daily difference between our FFD subsets was analysed by unpaired Welch two sample t-tests and statistical significance was defined as $p < 0.05$. The UK's annual mean FFD is the arithmetic mean of all observations available for each phenological year (i.e. between −100 DOY and 365 DOY). We used this simple measure in contrast to previous attempts [10], because it appears most robust for the period during which sample size is sufficiently high (i.e. from 1952 onwards). All histograms have 366 bins, and calculations were performed in R 4.0.2 [17], using the packages dplyr [18] and data.table [19]. The Central England Temperature series, obtained from the Met Office Hadley Centre (https://www.metoffice.gov.uk/hadobs/hadcet/data/download.html), was

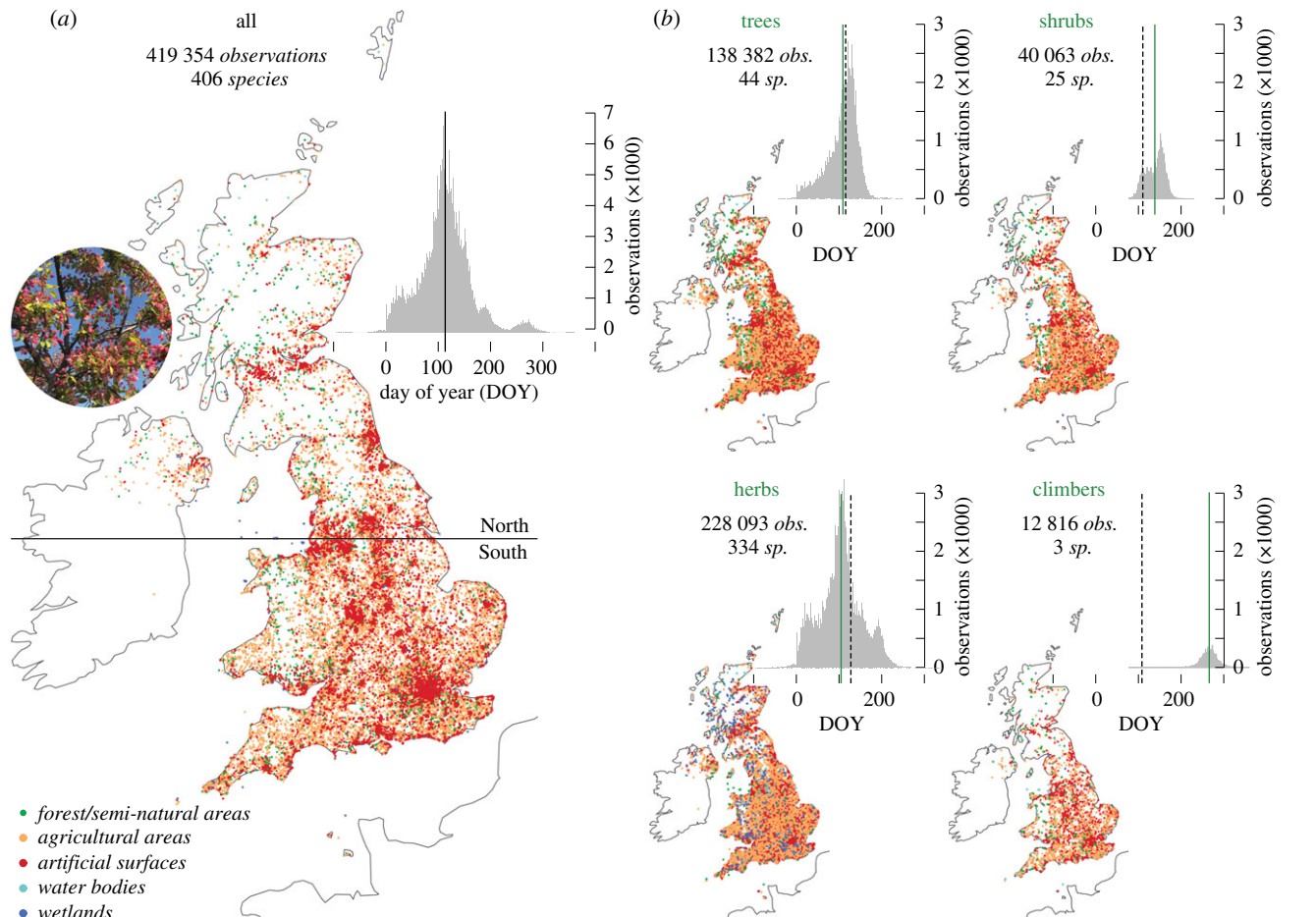

**Figure 1.** Network of first flowering dates. (*a*) Spatial distribution of 419 354 observations of daily resolved first flowering dates (FFDs) in 406 species between 1753 and 2019 across the UK. Dot colours refer to different landcover classifications based on the CORINE inventory, and the horizontal line at 53.5°N divides all obser-vations into 'North' and 'South' (87 247 and 332 107 FFDs). The histogram shows the intra-annual distribution of all FFDs, with the vertical black line representing the mean FFD (113 DOY). The inset photo shows the first flowering of a Japanese crab apple (*Malus floribunda*) on 26 April 2021, in Shudy Camps, South Cam-bridgeshire. (*b*) Spatial and intra-annual distribution of all FFD observations of trees, shrubs, herbs and climbers. The green vertical lines represent the lifeform specific mean FFDs, and the dashed lines are the mean FFD of the remaining data. (Online version in colour.)

used to assess changes in monthly, seasonal and annual tempera-ture means since 1659 CE (electronic supplementary material, figure S1). Monthly resolved, 0.25° × 0.25° gridded minimum, mean and maximum temperatures, precipitation totals and sea-level pressure indices, averaged over the British Isles between 50–60°N and 8°W and 2°E, and obtained from E-OBS v. 23.1 [20], were employed for comparison against the phenological data from 1952–2019, the period across which the UK-wide FFD dataset is most robust in terms of sample size (electronic supplementary material, figure S2). Monthly resolved indices of the North Atlantic Oscillation (NAO) derived from pressure differences between Gibraltar and Iceland for the years 1952–2019 [21] were used for additional comparisons. All climate data have been extracted from the KNMI climate explorer (http://climexp.knmi.nl/start.cgi?someone@somewhere).

## 3. Results

The UK's phenological record is dominated by 334 herb species, followed by 44 tree and 25 shrub species, whereas three different climber species account for only 0.7%. The spatial distribution of all FFDs largely corresponds to the human population distribution across the UK (figure 1*a*), with fewer records in the remote corners of the north and west. When the phenological observations are corrected for

species-specific extremes in the recorded first flowering time, the intra-annual distribution of FFDs, expressed as day of the year (DOY), range from mid-September of the pre-vious year to the end of December in the current year. The mean FFD of all observations is 23 April (113 DOY). Though the geographical coverage of the four lifeforms is quite similar (figure 1*b*), their mean FFDs vary from 103 to 266 DOY, with herbs and trees generally flowering first in mid-April, followed by shrubs a month later, and the few recorded climbers usually not flowering before September (table 1). Temporal gaps in the phenological observations affect the dataset in the years 1766, 1813/14 and 1817 (elec-tronic supplementary material, figure S2). The number of FFDs increases from 589 to 3061 in the years between 1891 and 1947, fluctuates from 118 to 893 between the years 1948 and 1998, and rises again since the year 1999 with a maxi-mum of 30 161 observations in the year 2007. While temporal changes in the number of sites are closely related to the number of observations, the number of species is high-est between the years 1952 and 1998. When dividing the complete UK's phenological dataset into two periods of older and more recent observations less than or equal to 1986 and greater than or equal to 1987 (figure 2), their respective mean FFDs are 132 and 106 DOY (table 1). This

**Table 1.** UK's first flowering dates. Number of observations and species (Obs. and Sp.), as well as the mean and standard error (S.E.) of the first flowering dates (FFDs) expressed as day of the year (DOY). Values partitioned by lifeform (trees, shrubs, herbs and climbers), a subset of 25 species that have been recorded continuously for at least 30 years in both split periods less than or equal to 1986 and greater than or equal to 1987, as well as biogeography (northern and southern UK, rural and urban settings as well as below and above 82 m sea level), are calculated for the full period and separately for the years to 1986 (older), and from 1987 (more recent). The difference in days (Diff.) between the older and more recent split periods is always significant ($p < 0.001$).

| | full period 1753–2019 | | | | older period ≤ 1986 | | | | recent period ≥ 1987 | | | | |
|---|---|---|---|---|---|---|---|---|---|---|---|---|---|
| | Obs. | Sp. | mean | S.E. | Obs. | Sp. | mean | S.E. | Obs. | Sp. | mean | S.E. | Diff. |
| all | 419 354 | 406 | 113 | 0.08 | 122 574 | 406 | 132 | 0.17 | 296 780 | 401 | 106 | 0.08 | 25.94 |
| trees | 138 382 | 44 | 109 | 0.09 | 32 717 | 44 | 120 | 0.16 | 105 665 | 42 | 106 | 0.11 | 14.76 |
| shrubs | 40 063 | 25 | 139 | 0.11 | 12 136 | 25 | 146 | 0.24 | 27 927 | 25 | 136 | 0.12 | 9.78 |
| herbs | 228 093 | 334 | 103 | 0.10 | 72 365 | 334 | 125 | 0.22 | 155 728 | 331 | 93 | 0.10 | 31.54 |
| climbers | 12 816 | 3 | 266 | 0.19 | 5356 | 3 | 268 | 0.28 | 7460 | 3 | 265 | 0.26 | 3.17 |
| 25 species | 369 166 | 25 | 109 | 0.08 | 78 068 | 25 | 122 | 0.19 | 291 098 | 25 | 105 | 0.08 | 16.92 |
| north | 87 247 | 73 | 118 | 0.17 | 26 754 | 68 | 139 | 0.35 | 60 493 | 61 | 110 | 0.19 | 29.08 |
| south | 332 107 | 406 | 112 | 0.09 | 95 820 | 406 | 130 | 0.19 | 236 287 | 401 | 105 | 0.09 | 24.98 |
| rural | 179 897 | 406 | 116 | 0.12 | 61 184 | 404 | 134 | 0.23 | 118 713 | 399 | 107 | 0.13 | 26.13 |
| urban | 232 291 | 313 | 111 | 0.11 | 58 749 | 307 | 130 | 0.24 | 173 542 | 125 | 105 | 0.11 | 25.14 |
| high | 168 039 | 406 | 114 | 0.12 | 48 958 | 406 | 133 | 0.25 | 119 081 | 395 | 106 | 0.13 | 26.90 |
| low | 251 315 | 247 | 113 | 0.10 | 73 616 | 202 | 131 | 0.22 | 177 699 | 215 | 106 | 0.11 | 25.31 |

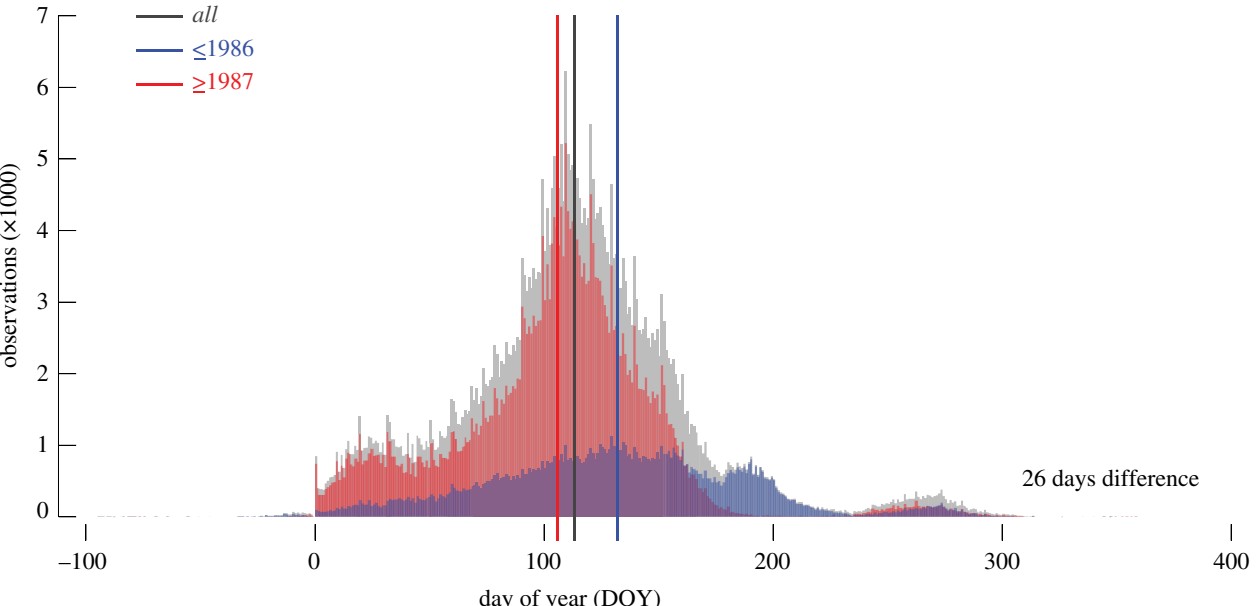

**Figure 2.** Temporal shifts in first flowering dates. Intra-annual distribution of all 419 354 first flowering dates (FFDs) between 1753 and 2019 (grey), together with their distribution after splitting into older (122 574 observations from 1753–1986) and more recent (296 780 observations from 1987–2019) FFDs. The difference of 26 days between the older and more recent FFDs is highly significant ($p < 0.0001$). (Online version in colour.)

difference of 26 days between the older (less than or equal to 1986) and more recent (greater than or equal to 1987) FFD subsets is highly significant ($p < 0.0001$).

The extent of the UK's phenological network allows the formation of well-replicated (bio)geographic subsets (figure 3): northern and southern sites (87 247 and 332 107 FFDs), urban and rural settings (232 291 and 179 897 FFDs) and lower and higher elevations (251 315 and 168 039 FFDs). The mean northern and southern FFDs are 118 and 112 DOY. The mean urban and rural FFDs are 111 and 116 DOY. The mean lower and higher elevation FFDs are 113 and 114 DOY. When further splitting the three (bio)geographic subsets into older and more recent periods (less than or equal to 1986 and greater than or equal to 1987), the mean northern FFDs are 139 and 110 DOY, and the mean southern FFDs are 130 and 105 DOY (figure 3a). The older and more recent urban mean FFDs are 130 and 105 DOY (figure 3b), and the older and more recent rural mean FFDs are 134 and 107 DOY. The older and more recent lower elevation mean FFDs are 131 and 106 DOY (figure 3c), and the older and more recent higher elevation mean FFDs are 133 and 106 DOY. The largest lifeform-specific difference of 32 days is found between 72 365 older and 155 728 more recent FFD observations in herbs (table 1), whereas smaller, though still significant ($p < 0.001$) differences are exhibited by trees and shrubs. Despite the relatively low sample sizes associated with further subdividing the UK's phenological dataset, a significant ($p < 0.001$) difference of 17 days is evident between 78 068 older and 291 098 more recent FFDs of those 25 species that have been recorded continuously for at least 30 years in both split periods less than or equal to 1986 and greater than or equal to 1987 (figure 4a; table 1). Independent of temporal changes, we also found a small difference of two days between 26 091 observations of 55 annual herb species and 200 341 observations of 351 perennial herb species (figure 4b), as well as a massive difference of 41 days between 74 wind-pollinated herb species and 235 insect-pollinated herb species (figure 4c), for which 17 673 and 209 295 observations exist respectively.

We consider the UK's mean FFDs most reliable between 1952 and 2019 when the numbers of observations and species is highest (electronic supplementary material, figure S2). During this period, our mean FFD record correlates at $r = 0.81$ ($p < 0.0001$) with the first flowering index of Amano et al. [10]. During the first 34 years from 1952–1985, our mean FFD is 139 DOY, but drops sharply afterward to 106 DOY (figure 5a). The largest year-to-year mean difference of more than two months is found between 7 June 1968 and 2 April 2019. Mean FFDs from 1952–2019 have a highly negative correlation with January–April maximum temperatures over the British Isles ($r = -0.81$; $p < 0.0001$; figure 5b). Almost identical correlations of $-0.78$ and $-0.73$ are obtained between the mean FFD and mean and minimum temperatures, respectively (electronic supplementary material, figure S3). At the monthly level, February maximum, mean and minimum temperatures have the strongest negative association with FFDs, whereas January, March and April monthly temperatures have weaker correlations with the phenological record. The phenological observations and temperature measurements jointly exhibit a shift associated with the change from a generally negative to an overall positive mode of the NAO in the second half of the 1980s that coincides with a temperature increase of approximately 1°C (figure 5b). Spatial field correlations between the UK's community-wide record of mean FFDs and gridded global January–April temperatures are not only highly significant over western Europe ($p < 0.0001$), but also over many regions in both hemispheres that likely experienced similar warming rates since the mid-twentieth century (figure 5c). The observed year-to-year covariance between the first-differenced phenological and temperature data is, however, largely restricted to the British Isles, as well as parts of France, the Benelux countries and southern Scandinavia. While interannual and longer-term changes in the UK's FFDs are highly sensitive to January–April temperatures, neither the sufficiently high precipitation totals in winter and spring over the British Isles, nor daylight length play an important role in year-to-year changes in first flowering dates, and thus the onset of the growing season.

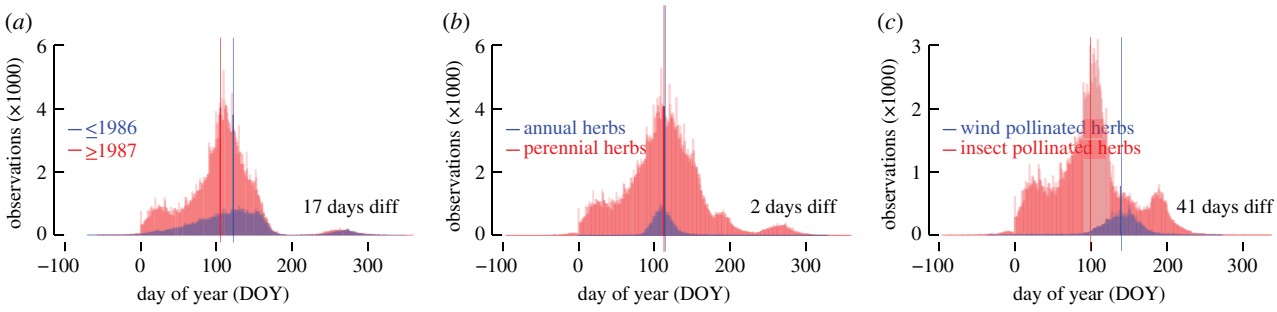

**Figure 3.** Site effects on first flowering dates. (*a*) Intra-annual distribution of 87 247 northern and 332 107 southern FFDs between 1753 and 2019, together with their distribution after splitting into older (1753–1986) and more recent (1987–2019) FFDs. (*b*) Intra-annual distribution of 179 897 rural and 232 291 urban FFDs between 1753 and 2019, together with their distribution after splitting into older (1753–1986) and more recent (1987–2019) FFDs. (*c*) Intra-annual distribution of 168 039 high-elevation and 251 315 low-elevation FFDs between 1753 and 2019, together with their distribution after splitting into older (1753–1986) and more recent (1987–2019) FFDs. All differences between the older and more recent FFDs are highly significant (*p* < 0.001). (Online version in colour.)

**Figure 4.** Robustness of first flowering dates. (*a*) Intra-annual distribution of 78 068 older (1753–1986) and 291 098 more recent (1987–2019) first flowering dates (FFDs) of the 25 species that have been recorded continuously for at least 30 years in both split periods less than or equal to 1986 and greater than or equal to 1987 (*Aesculus hippocastanum, Alliaria petiolata, Alopecurus pratensis, Anemone nemorosa, Betula pendula, Cardamine pratensis, Corylus avellana, Crataegus monogyna, Dactylis glomerata, Ficaria verna, Galanthus nivalis, Hedera helix, Holcus lanatus, Hyacinthoides non-scripta, Leucanthemum vulgare, Narcissus pseudonarcissus, Phleum pratense, Prunus spinosa, Quercus petraea, Quercus robur, Rosa canina, Sambucus nigra, Sorbus aucuparia, Syringa vulgaris* and *Tussilago farfara*). (*b*) Intra-annual distribution of 26 091 observations of 55 annual herb species and 200 341 observations of 351 perennial herb species between 1753 and 2019. (*c*) Intra-annual distribution of 17 673 observations of 74 wind-pollinated herb species and 209 295 observations of 235 insect-pollinated herb species between 1753 and 2019. (Online version in colour.)

## 4. Discussion

Our study reveals that the UK's community-wide mean FFDs advanced by almost one month from the mid-1980s compared to all phenological observations of the preceding years since 1753 CE. Furthermore, we show that many plants in the UK are flowering almost one week earlier in the south than in the north, at lower than higher elevations,

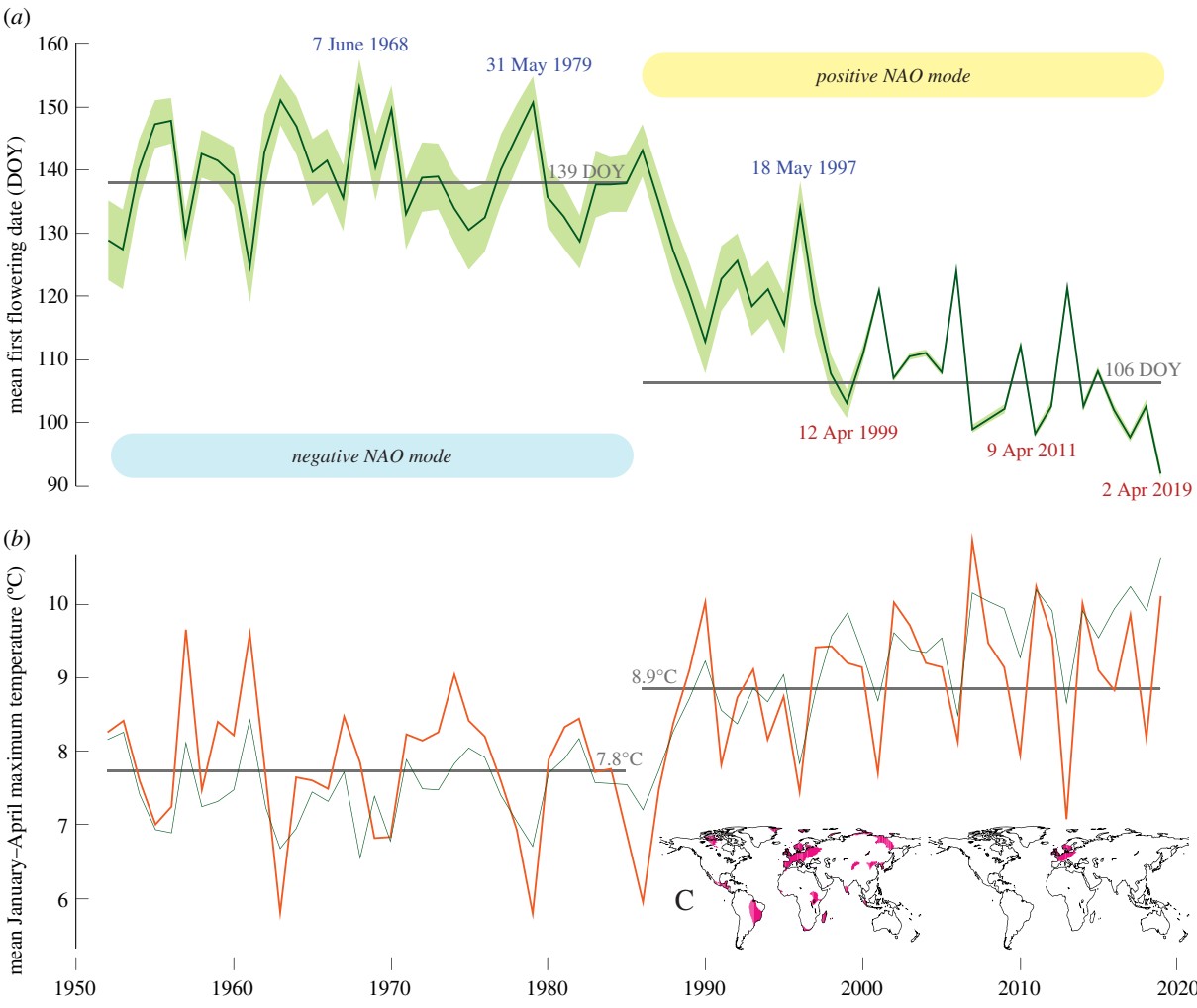

**Figure 5.** Temperature dependency of first flowering dates. (a) The UK's mean first flowering dates (FFDs) between 1952 and 2019 (green line), together with its 95% confidence interval (shading). Extreme years of early and late FFDs are marked and detailed, and the horizontal grey lines refer to the mean FFDs of two equally long, 34-year split periods from 1952–1985 and 1986–2019 and have been calculated from all observations (table 1). Two phases during which negative and positive modes of North Atlantic Oscillation (NAO) dominated are indicated. (b) Mean January–April maximum temperature averaged over the British Isles between 50–60° north and 8° west and 2° east (orange), together with the inverse, scaled record of mean FFDs (green). (c) Pink areas refer to highly significant ($p < 0.0001$) spatial field correlations between the spring-dominated UK's mean FFDs and gridded mean January–April maximum temperatures using the original and first-difference time series from 1952–2019 (left and right maps). (Online version in colour.)

and under urban compared to rural settings. The observed trend in an earlier first flowering time during the past three decades is most pronounced in herbs. While phenotypic plasticity is more important for long-lived trees and shrubs, directional selection and adaptive evolution can be faster in short-lived plants that exhibit faster turnover rates [22]. However, we do not know whether adaptive evolution will allow populations to reach new phenotypic optima rapidly enough to keep pace with climate change [23]. Interannual differences in the observed phenophase of all plants in the UK can be best explained by mean January–April maximum temperatures over the British Isles, because the amount of precipitation and the timing of snowmelt are not important for the first flowering at most observational sites. A distinct shift in both the FFDs and spring warming since the mid-twentieth century corresponded with an abrupt change of the NAO mode in the second half of the 1980s. Observational evidence for a regime shift in the aftermath of the El Chichón volcanic eruption in 1982 has been reported from different parts of the world [24], indicating the relevance of such events to large-scale climate dynamics, rapid environmental responses and cascading trophic interactions.

Although the geographical (north–south, low–high and urban–rural) and temporal (before and after 1986) differences in the UK's FFDs are highly significant ($p < 0.001$), caution is advised since there are substantial differences in the number of observations, species and sites over space and time (electronic supplementary material, figure S2). To avoid overinterpretation of noisy timeseries behaviour before the mid-twentieth century when data quality and quantity is relatively low, we restricted most analyses to the 1952–2019 period during which both the phenological observations and the climatic measurements are most reliable. In addition to the record's temporal heterogeneity, there are fewer observations in the north than in the south (table 1), at higher than lower elevations, and for shrubs and climbers compared to herbs and trees. Further complexity comes from irrevocable land-use/land-cover changes since the mid-eighteenth century, which were particularly extensive in areas of early industrialization. Ongoing urban sprawl and the associated impact of heat islands challenge the assessment of temporal changes in mean FFDs of our urban and rural subsets. Possible recording errors and different perceptions of individual observers, spatio-temporal changes in species composition

and observer location, as well as ontogenetic effects arising from the long-term monitoring of individual plants, cannot be disregarded.

Species-specific mean FFDs are widely distributed from late December of the previous year (*H. foetidus*; −7.3 DOY) to late September of the current year (*Hedera helix*; 266.9 DOY) (electronic supplementary material, figure S4). On the level of individual observations, the range of FFDs stretches from the previous autumn until the end of the current year (from −100 DOY to 365 DOY), suggesting that photoperiodism (the ratio between day and night length) has not yet affected our results. However, if temperatures continue to rise and further shift the mean FFDs from April into March or even earlier, then light availability that controls the formation of winter buds, leaf abscission and freezing resistance [25,26], could become a critical factor for bud break and first flowering of many plant species [27]. The photoperiod threshold will be more important for long-lived, late-successional species that generally flower early in the year and grow at high latitudes [28]. Rather than understanding photoperiodism as a constraint, it should be considered as a safeguard that protects plants from frost damage, synchronizes their development stages, and facilitates trophic interaction between different organisms [9]. This is important since winter warming has exceeded the temperature increase of other months [9], with implications not only for the amplitude of the annual climate cycle [29] but also for the alignment between biological requirements of different ecosystem components [30,31]. Earlier spring bloom will increase the probability of frost damage in natural and agricultural systems [7]. A prolonged flowering period will shift and extend the season of respiratory allergies associated with plant pollen [11,12]. Though varying between organisms and habitats, trends and extremes in spring climate may alter species-specific chilling requirements, the likelihood of frost injury and demands on energy and water balance [8]. The timing of plant flowering can affect their pollination, especially when insect pollinators are themselves seasonal, and determine the timing of seed ripening and dispersal. Plant flowering also influences animals for which pollen, nectar, fruits and seeds are important resources, and earlier flowering implies earlier activity in leaf expansion, root growth and nutrient uptake, which are important for niche differentiation among coexisting species. Large changes in FFDs are therefore expected to disrupt community composition and interaction [9]. Ecological mismatch can increase extinction risks and loss of ecosystem services [32,33]. In addition to temporal shifts in the onset of growing seasons will their prolongation enhance the capacity to assimilate carbon dioxide from the atmosphere. Further to mitigating the effects of greenhouse gases [34,35], growing net primary productivity will affect carbon cycles and budgets from local to global scales [8,36,37].

Based on valuable citizen science and exceeding previous findings at an alarming rate, our study reveals an average phenological advancement of 5.4 days per decade between 1952 and 2019. The observed maximum range of annual mean FFDs in this period is more than two months (7 June 1968 versus 2 April 2019). Mean flowering time is one day earlier at lower than higher elevations, 6 days earlier in the south than in the north, and 5 days earlier under urban than rural conditions. From 1952–2019 the phenological record correlates with January–April maximum temperatures at $r = -0.81$ ($p < 0.0001$). Both, the phenological and climatological data reflect an abrupt shift from an overall negative to a more positive mode of the NAO in the second half of the 1980s. Beneficial aspects of earlier FFDs are likely to vanish if planet Earth continues to warm and ecological mismatch kicks in. Notably, our observed phenological trends and extremes are much greater than those reported by the UK Spring Index that informs the British government and is used for public guidance [38]. This record, however, only includes the first flowering time of two species, hawthorn (*C. monogyna*) and horse chestnut (*Aesculus hippocastanum*), both of which have a rate of advancement lower, by roughly 15 days, than the community-wide advancement in the UK's mean FFDs of our study (electronic supplementary material, figure S5). Independent of the data used and methods applied, we conclude that if plants in the UK continue to flower earlier, and if the frequency, intensity and duration of climatic extremes increase further, the functioning and productivity of biological, ecological and agricultural systems will be at an unprecedented risk.

**Ethics.** The authors declare no ethical issues.

**Data accessibility.** The phenological dataset is available from the UK Nature's Calendar of the Woodland Trust: https://naturescalendar.woodlandtrust.org.uk/.

**Authors' contributions.** U.B.: conceptualization, formal analysis, investigation, project administration, writing—original draft; A.P.: data curation, formal analysis, writing—original draft; P.J.K.: formal analysis, writing—original draft; J.E.: writing—original draft; T.S.: writing—original draft; A.C.: data curation, formal analysis, writing—original draft.

All authors gave final approval for publication and agreed to be held accountable for the work performed therein.

**Competing interests.** The authors declare no competing interests.

**Funding.** U.B and J.E. received funding from SustES: Adaptation strategies for sustainable ecosystem services and food security under adverse environmental conditions (CZ.02.1.01/0.0/0.0/16_019/0000797), and the ERC project MONOSTAR (AdG 882727). A.C. was supported by the Fritz and Elisabeth Schweingruber Foundation, and the Woodland Trust.

**Acknowledgements.** Phenological data were obtained from the UK Nature's Calendar (Woodland Trust), and we are particularly grateful to all citizen scientists who added observations. Lorienne Whittle from the Woodland Trust stimulated discussion and supported data selection.

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
