## [Peer Review File · Proceedings of the Royal Society B: Biological Sciences]

Review History

RSPB-2021-1878.R0 (Original submission)

Review form: Reviewer 1

Recommendation

Major revision is needed (please make suggestions in comments)

Scientific importance: Is the manuscript an original and important contribution to its field?

Good

General interest: Is the paper of sufficient general interest?

Good

Quality of the paper: Is the overall quality of the paper suitable?

Good

Is the length of the paper justified?

Yes

Should the paper be seen by a specialist statistical reviewer?

Yes

Do you have any concerns about statistical analyses in this paper? If so, please specify them explicitly in your report.

Yes

It is a condition of publication that authors make their supporting data, code and materials available - either as supplementary material or hosted in an external repository. Please rate, if applicable, the supporting data on the following criteria.

Is it accessible?

Yes

Is it clear?

Yes

Is it adequate?

Yes

Do you have any ethical concerns with this paper?

No

Comments to the Author

Using 419,354 observations of the first flowering date (FFT) in UK, this manuscript described the recent trends of FFT, and explored the temperature driver of FFT. The earlier start of growing season (SOS) under the global warming has already been reported in previous finding, however, FFT has not been much concerned, since PPT is difficult to be identified by remote sensing and PPT is as closely related to GPP as SOS or EOS. I support that PPT is also worth studying, and physiological record, such as PEP725 and Nature's Calendar, provide a valuable data source for PPT research. The subject is relevant and the scientific questions are addressed with a robust dataset. The structure is complete and reasonable, results are clearly and concisely presented. I am recommending acceptance after several concerns suggestions being addressed during the revision.

Main concerns

Abstract

So short and not clear enough.

The relationship between temperature and PPT has not been presented in this part.

'Genetic adaptation' has not been strictly proved in manuscript.

Introduction

Please put more emphasis on the necessity of studying PPT, what previous studies have done, the research gap, and the innovation of this paper.

Method

The analysis strategy has not been mentioned. When explore the correlations between PPT and temperature, other climatic factors, such as precipitation and photoperiod, also have a compound impact. I recommend using partial correlation coefficient rather than Pearson's correlation to eliminate the interfere of other factors.

Result

I think the division between north or south, rural or urban, high or low is too rigid. Why not use latitude, distance from city center, and Altitude as coordinate axis, which will depict more detailed information.

The pre-season before PPT may be consider then the fixed month in Figure S6.

Except maximum, mean and minimum temperatures, you can try the CV of temperature.

Discussion

The discussion section does not delve deep enough. The discussion could be improved by emphasizing the final messages.

Review form: Reviewer 2

Recommendation

Accept with minor revision (please list in comments)

Scientific importance: Is the manuscript an original and important contribution to its field?

Good

General interest: Is the paper of sufficient general interest?

Good

Quality of the paper: Is the overall quality of the paper suitable?

Acceptable

Is the length of the paper justified?

Yes

Should the paper be seen by a specialist statistical reviewer?

No

Do you have any concerns about statistical analyses in this paper? If so, please specify them explicitly in your report.

Yes

It is a condition of publication that authors make their supporting data, code and materials available - either as supplementary material or hosted in an external repository. Please rate, if applicable, the supporting data on the following criteria.

Is it accessible?

No

Is it clear?

No

Is it adequate?

No

Do you have any ethical concerns with this paper?

No

Comments to the Author

General comments

The authors investigated the first flowering dates using a large dataset in UK, over a quiet long time period 1753 - 2019, and they found one month earlier after 1986, and earlier flowering at low altitude and latitudes, as well as in urban area. This is interesting study, and valuable using such impressive dataset. Climate warming advanced spring phenology has been well documented, as well its spatial difference, i.e. along latitude and altitudes. I'm only one question about the methodology, which may damage the results.

The authors found and highlighted that the first flowering dates were 26 days difference between 1735-1986 and 1987-2019, as well as some days difference between high and low altitudes and latitudes, that's interesting, but might be large uncertainty in these estimations, because of larger difference in species composition, phenological sites and records numbers between these two time periods. Although the authors mentioned this limitation in the discussion, but too simple, and need further investigation. To test the robustness, it would be necessary to check these difference at species level, in details, the authors may estimate the diff of FFDs for each species at each site, and then calculate the mean +/- std values. In this way, the estimation should be more reliable.

Specific comments

L81-83, how many sites were selected?

L103-104, why chose 53.5oN as the separate point between southern and northern? same as the 150m as the point between lower and higher elevation?

L112, the resolution of the WorldClim should be provided.

L132-134, the figures are replicated with above, here may remove this background information.

L210-213, other limitation need to be discussed, for example the ontogenetic difference, as well as the monitored trees, shrubs as well as plant were likely different during different time periods.

L221-222, yes, the direct experimental proof, see Fu et al., 2019 gcb, they reported a photoperiod effect only when the warming treatment after a threshold.

Short photoperiod reduces the temperature sensitivity of leaf-out in saplings of *Fagus sylvatica* but not in horse chestnut. *Global change biology*, 2019,

L246, 5.4 days per decade, provide the time period, I guess it should be 1952-2019, or even late?

L258-259, the argument is unclear. a community wide shifts in flowering time will be harmful for biology, ecology and society, more details need to be provided.

Decision letter (RSPB-2021-1820.R0)

05-Oct-2021

Dear Dr Büntgen:

I am writing to inform you that your manuscript RSPB-2021-1820 entitled "Plants in the UK flower a month earlier under recent warming" has, in its current form, been rejected for publication in *Proceedings B*.

This action has been taken on the advice of referees, who have recommended that substantial revisions are necessary. With this in mind we would be happy to consider a resubmission, provided the comments of the referees are fully addressed. However please note that this is not a provisional acceptance.

1) A 'response to referees' document including details of how you have responded to the comments, and the adjustments you have made.

- 2) A clean copy of the manuscript and one with 'tracked changes' indicating your 'response to referees' comments document.
- 3) Line numbers in your main document.
- 4) Data - please see our policies on data sharing to ensure that you are complying (<https://royalsociety.org/journals/authors/author-guidelines/#data>).

Sincerely,
 Dr Locke Rowe
 mailto: proceedingsb@royalsociety.org

Associate Editor
 Board Member: 1
 Comments to Author:

Using a big amount of observations of first flower date (FFD) data, the authors of this manuscript evaluated how FFD changed over two time periods, i.e. before 1986 and after 1986. They found that on average the FFD of all studied species changed from 132 days in the earlier period to 106 days in the more recent period, leading to a significant earlier FFD in UK. This trend was consistent between different lifeforms, different regions and different elevations, although the number of days of FFD change differed. The changes in FFD was likely determined by the mean maximum T of January-April. Two reviewers have reviewed the manuscript. Although both of them are positive on the topic of this manuscript, they also both raised severe concerns about the methodology and writing. I highly agree with their comments on the analytical methods used here. I think the authors should carefully deal with all these comments raised by the reviewers. In addition, I have a couple concerns about both methods and results. First, the observations were conducted at different sites in different years. As different regions (e.g. different latitude, altitude, and distance to the urban areas) may have different FFD, changes in observation sites, especially a changing trend in observation sites, may significantly bias the observed changes in FFD. For example, if the more recent observations are more concentrated in urban areas than the older observations, this may lead to a larger decrease in FFD than expected. The potential influence of the changes in observation sites on the results should be clearly dealt with in the statistical models. Second, a clear peak of FFD in the more recent period is observed, but not for the earlier period. This may suggest that the species composition has significantly changed. The reason for the difference between the frequency distributions during the two periods and the consequences of this difference on the results should be more clearly explored. Moreover, the herb species with only one or two records may bias the results. Should these records be included in the analyses? Lines 130 - 134 are repeating the methods.

Reviewer(s)' Comments to Author:
 Referee: 1
 Comments to the Author(s)

Using 419,354 observations of the first flowering date (FFT) in UK, this manuscript described the recent trends of FFT, and explored the temperature driver of FFT. The earlier start of growing season (SOS) under the global warming has already been reported in previous finding, however, FFT has not been much concerned, since PPT is difficult to be identified by remote sensing and PPT is as closely related to GPP as SOS or EOS. I support that PPT is also worth studying, and physiological record, such as PEP725 and Nature's Calendar, provide a valuable data source for PPT research. The subject is relevant and the scientific questions are addressed with a robust dataset. The structure is complete and reasonable, results are clearly and concisely presented. I

am recommending acceptance after several concerns suggestions being addressed during the revision.

Main concerns

Abstract

So short and not clear enough.

The relationship between temperature and PPT has not been presented in this part.

‘Genetic adaptation’ has not been strictly proved in manuscript.

Introduction

Please put more emphasis on the necessity of studying PPT, what previous studies have done, the research gap, and the innovation of this paper.

Method

The analysis strategy has not been mentioned. When explore the correlations between PPT and temperature, other climatic factors, such as precipitation and photoperiod, also have a compound impact. I recommend using partial correlation coefficient rather than Pearson’s correlation to eliminate the interfere of other factors.

Result

I think the division between north or south, rural or urban, high or low is too rigid. Why not use latitude, distance from city center, and Altitude as coordinate axis, which will depict more detailed information.

The pre-season before PPT may be consider then the fixed month in Figure S6.

Except maximum, mean and minimum temperatures, you can try the CV of temperature.

Discussion

The discussion section does not delve deep enough. The discussion could be improved by emphasizing the final messages.

Referee: 2

Comments to the Author(s)

General comments

The authors investigated the first flowering dates using a large dataset in UK, over a quiet long time period 1753 – 2019, and they found one month earlier after 1986, and earlier flowering at low altitude and latitudes, as well as in urban area. This is interesting study, and valuable using such impressive dataset. Climate warming advanced spring phenology has been well documented, as well its spatial difference, i.e. along latitude and altitudes. I’m only one question about the methodology, which may damage the results.

The authors found and highlighted that the first flowering dates were 26 days difference between 1735-1986 and 1987-2019, as well as some days difference between high and low altitudes and latitudes, that's interesting, but might be large uncertainty in these estimations, because of larger difference in species composition, phenological sites and records numbers between these two time periods. Although the authors mentioned this limitation in the discussion, but too simple, and need further investigation. To test the robustness, it would be necessary to check these difference at species level, in details, the authors may estimate the diff of FFDs for each species at each site, and then calculate the mean +- std values. In this way, the estimation should be more reliable.

Specific comments

L81-83, how many sites were selected?

L103-104, why chose 53.5oN as the separate point between southern and northern? same as the 150m as the point between lower and higher elevation?

L112, the resolution of the WorldClim should be provided.

L132-134, the figures are replicated with above, here may remove this background information.
 L210-213, other limitation need to be discussed, for example the ontogenetic difference, as well as the monitored trees, shrubs as well as plant were likely different during different time periods.
 L221-222, yes, the direct experimental proof, see Fu et al., 2019 gcb, they reported a photoperiod effect only when the warming treatment after a threshold.
 Short photoperiod reduces the temperature sensitivity of leaf-out in saplings of *Fagus sylvatica* but not in horse chestnut. *Global change biology*, 2019,
 L246, 5.4 days per decade, provide the time period, I guess it should be 1952-2019, or even late?
 L258-259, the argument is unclear. a community wide shifts in flowering time will be harmful for biology, ecology and society, more details need to be provided.

Author's Response to Decision Letter for (RSPB-2021-1820.R0)

See Appendix A.

RSPB-2021-2456.R0

Review form: Reviewer 1

Recommendation

Accept as is

Scientific importance: Is the manuscript an original and important contribution to its field?

Good

General interest: Is the paper of sufficient general interest?

Good

Quality of the paper: Is the overall quality of the paper suitable?

Good

Is the length of the paper justified?

Yes

Should the paper be seen by a specialist statistical reviewer?

No

Do you have any concerns about statistical analyses in this paper? If so, please specify them explicitly in your report.

No

It is a condition of publication that authors make their supporting data, code and materials available - either as supplementary material or hosted in an external repository. Please rate, if applicable, the supporting data on the following criteria.

Is it accessible?

Yes

Is it clear?

Yes

Is it adequate?

Yes

Do you have any ethical concerns with this paper?

No

Comments to the Author

I think authors have done a good work in the revision and all my previous comments and concerns were well addressed. It is now to accept the manuscript.

Review form: Reviewer 2

Recommendation

Accept with minor revision (please list in comments)

Scientific importance: Is the manuscript an original and important contribution to its field?

Good

General interest: Is the paper of sufficient general interest?

Good

Quality of the paper: Is the overall quality of the paper suitable?

Acceptable

Is the length of the paper justified?

Yes

Should the paper be seen by a specialist statistical reviewer?

Yes

Do you have any concerns about statistical analyses in this paper? If so, please specify them explicitly in your report.

Yes

It is a condition of publication that authors make their supporting data, code and materials available - either as supplementary material or hosted in an external repository. Please rate, if applicable, the supporting data on the following criteria.

Is it accessible?

N/A

Is it clear?

N/A

Is it adequate?

N/A

Do you have any ethical concerns with this paper?

No

Comments to the Author

Although the manuscript was improved, but the major concern was not solved. The robustness need to be tested as I mentioned, because of different species composition, different sites and

different regions were used between these two time periods, see my previous review. I would suggest the authors to do further test, for example as I suggested in my review, calculate the mean \pm std values at species level, and then compare its different between different time periods.. you may also check the difference at species level, or using same species between these two periods..

by the way, the authors only provided the reason why chose 53.5oN as the separate point between southern and northern? why using the 150m as the point between lower and higher elevation? it's not clear. please reply to each comment.

Furthermore, I agree with the reviewer#1, a partial correlation analysis, rather simple Pearson correlation is more suitable, and need to be tested as well.

Decision letter (RSPB-2021-2456.R0)

24-Dec-2021

Dear Dr Büntgen

I am pleased to inform you that your manuscript RSPB-2021-2456 entitled "Plants in the UK flower a month earlier under recent warming" has been accepted for publication in Proceedings B.

The referee's comments are mixed, but my reading of them and the MS suggests some minor revisions to your manuscript will be sufficient. Therefore, I invite you to respond to the comments and revise your manuscript. Because the schedule for publication is very tight, it is a condition of publication that you submit the revised version of your manuscript within 7 days. If you do not think you will be able to meet this date please let us know.

- 1) A text file of the manuscript (doc, txt, rtf or tex), including the references, tables (including captions) and figure captions. Please remove any tracked changes from the text before submission. PDF files are not an accepted format for the "Main Document".
- 2) A separate electronic file of each figure (tiff, EPS or print-quality PDF preferred). The format should be produced directly from original creation package, or original software format. PowerPoint files are not accepted.

3) Electronic supplementary material: this should be contained in a separate file and where possible, all ESM should be combined into a single file. All supplementary materials accompanying an accepted article will be treated as in their final form. They will be published alongside the paper on the journal website and posted on the online figshare repository. Files on figshare will be made available approximately one week before the accompanying article so that the supplementary material can be attributed a unique DOI.

Sincerely,

Dr Locke Rowe

Editor:

The main concern here is the heterogeneity in the before and after data sets. I think you have taken some steps in the discussion to emphasize this suggests some caution in interpreting your

findings. I would just like to see a little more emphasis on this in the discussion. On the upside, I think your point about S3 could be a good one and should be included in the discussion. I am assuming that "continuously" means each year and in the same sites?.

Reviewer(s)' Comments to Author:

Referee: 1

Comments to the Author(s).

I think authors have done a good work in the revision and all my previous comments and concerns were well addressed. It is now to accept the manuscript.

Referee: 2

Comments to the Author(s).

Although the manuscript was improved, but the major concern was not solved. The robustness need to be tested as I mentioned, because of different species composition, different sites and different regions were used between these two time periods, see my previous review. I would suggest the authors to do further test, for example as I suggested in my review, calculate the mean \pm std values at species level, and then compare its different between different time periods.. you may also check the difference at species level, or using same species between these two periods...

by the way, the authors only provided the reason why chose 53.50N as the separate point between southern and northern? why using the 150m as the point between lower and higher elevation? it's not clear. please reply to each comment.

Furthermore, I agree with the reviewer#1, a partial correlation analysis, rather simple Pearson correlation is more suitable, and need to be tested as well.

Author's Response to Decision Letter for (RSPB-2021-2456.R0)

See Appendix B.

Decision letter (RSPB-2021-2456.R1)

04-Jan-2022

Dear Dr Büntgen

I am pleased to inform you that your manuscript entitled "Plants in the UK flower a month earlier under recent warming" has been accepted for publication in Proceedings B.

Your article has been estimated as being 9 pages long. Our Production Office will be able to confirm the exact length at proof stage.

Data Accessibility section

Open Access

Paper charges

Sincerely,

Proceedings B

Appendix A

UNIVERSITY OF
CAMBRIDGE

Department of Geography

Professor Ulf Büntgen

Professor of Environmental Systems Analysis

Prof Locke Rowe
Editor *Proceedings of the Royal Society B*
Earth Sciences Centre, University of Toronto, Canada

10th November 2021

Dear Prof Rowe

Please find attached our carefully revised manuscript entitled '**Plants in the UK flower a month earlier under recent warming**' (RSPB-2021-1820), which we hope is now suitable for publication in *Proceedings B*.

We are thankful to both referees and the Associate Editor for their helpful reviews. All of them provided constructive and encouraging comments and suggestions, which we considered thoroughly during revision. All changes and subsequent improvements are detailed below in our point-by-point response letter.

On behalf of the authors

Yours sincerely

Ulf Büntgen

Associate Editor / Board Member:

Using a big amount of observations of first flower date (FFD) data, the authors of this manuscript evaluated how FFD changed over two time periods, i.e. before 1986 and after 1986. They found that on average the FFD of all studied species changed from 132 days in the earlier period to 106 days in the more recent period, leading to a significant earlier FFD in UK. This trend was consistent between different lifeforms, different regions and different elevations, although the number of days of FFD change differed. The changes in FFD was likely determined by the mean maximum T of January-April. Two reviewers have reviewed the manuscript. Although both of them are positive on the topic of this manuscript, they also both raised severe concerns about the methodology and writing. I highly agree with their comments on the analytical methods used here. I think the authors should carefully deal with all these comments raised by the reviewers. In addition, I have a couple concerns about both methods and results. First, the observations were conducted at different sites in different years. As different regions (e.g. different latitude, altitude, and distance to the urban areas) may have different FFD, changes in observation sites, especially a changing trend in observation sites,

Downing Place
Cambridge CB2 3EN
Tel: +44 (0) 1223 760564
Fax: +44 (0) 1223 333392
Email: ulf.buentgen@geog.cam.ac.uk

may significantly bias the observed changes in FFD. For example, if the more recent observations are more concentrated in urban areas than the older observations, this may lead to a larger decrease in FFD than expected. The potential influence of the changes in observation sites on the results should be clearly dealt with in the statistical models. Second, a clear peak of FFD in the more recent period is observed, but not for the earlier period. This may suggest that the species composition has significantly changed. The reason for the difference between the frequency distributions during the two periods and the consequences of this difference on the results should be more clearly explored. Moreover, the herb species with only one or two records may bias the results. Should these records be included in the analyses?

We are thankful for this thorough evaluation, have considered all comments and suggestions, and have improved the manuscript accordingly. Please note our 'biogeographic' data splitting (see table 1 for an overview together with figures S3–S5), as well as the rather conservative temporal restriction to the 1952–2019 period during which both the phenological observations and climatic measurements are most reliable. All possible measures have been performed and implemented to avoid bias in and overinterpretation of our findings. The observed ~26-day earlier flowering in UK plants, however, is a robust finding that is best reflected by using the entire collection of 419,354 first flowering dates (figure 2).

Please also note that the peak in FFD distribution most likely relates to sample size, because we always see a peak if the number of observations is high but lose this peak if the number of observations declines relatively.

Lines 130–134 are repeating the methods.

We re-wrote the section and removed all methodological description to avoid redundancy (see also our response to referee 2).

Referee 1:

Using 419,354 observations of the first flowering date (FFT) in UK, this manuscript described the recent trends of FFT, and explored the temperature driver of FFT. The earlier start of growing season (SOS) under the global warming has already been reported in previous finding, however, FFT has not been much concerned, since PPT is difficult to be identified by remote sensing and PPT is as closely related to GPP as SOS or EOS. I support that PPT is also worth studying, and physiological record, such as PEP725 and Nature's Calendar, provide a valuable data source for PPT research. The subject is relevant and the scientific questions are addressed with a robust dataset. The structure is complete and reasonable, results are clearly and concisely presented. I am recommending acceptance after several concerns suggestions being addressed during the revision.

We are thankful for this thorough evaluation, have considered all comments and suggestions, and have improved the manuscript accordingly. We assume that FFT and PPT both refer to the first flowering date, which is usually abbreviated as FFD.

Abstract

So short and not clear enough.

We expanded the abstract from 138 to 187 words. However, we also followed the journal-specific guidelines (“The abstract should be no more than 200 words and should not contain references or unexplained abbreviations or acronyms. Your abstract should be concise and informative and should read well as a standalone piece. The general scope of the article as well as the main results and conclusions should be summarised.”).

The relationship between temperature and PPT has not been presented in this part.

We added “Global temperatures are rising at an unprecedented rate in the context of Common Era climate variability, but environmental responses are often difficult to recognise and quantify. Long-term observations of plant phenology, the annually recurring sequence of plant developmental stages, can provide sensitive measures of climate change and important information for ecosystem services.”.

‘Genetic adaptation’ has not been strictly proved in manuscript.

We re-wrote the sentence “The largest phenological shift of 32 days is found in short-lived herbs characterised by fast turnover rates and potentially high levels of genetic adaptation.”.

Introduction

Please put more emphasis on the necessity of studying PPT, what previous studies have done, the research gap, and the innovation of this paper.

We expanded the ‘Introduction’ accordingly and added several details, including the following section “The scientific value of hundreds of thousands of observations of first flowering dates from a wide range of plant species is related to the high temperature sensitivity of this phenological event, the large sample size of citizen science datasets, and the high spatiotemporal precision of the observed intra- and interannual changes [see 8 for an extensive review]. However, such studies operating at a community-level and spanning longer timescales and larger biogeographic regions are particularly rare.”.

Please also note that the revised version of the manuscript describes the state-of-the-art and value of plant phenological data from lines 57–77.

Method

The analysis strategy has not been mentioned. When explore the correlations between PPT and temperature, other climatic factors, such as precipitation and photoperiod, also have a compound impact. I recommend using partial correlation coefficient rather than Pearson’s correlation to eliminate the interfere of other factors.

We actually did compare the FFDs against a range of climatic variables that change from year to year and wrote “Monthly-resolved, 0.25° x 0.25° gridded minimum, mean and maximum temperatures, precipitation totals, and sea level pressure, averaged over the British Isles between 50–60°N and 8°W and 2°E, from E-OBS v23.1 [23], were used for comparison against the phenological data from 1952–2019, the period across which the FFD data are the most robust in terms of mean and variance.”.

However, since photoperiod does not change at interannual timescales, spring precipitation totals across the UK are both sufficient for growth and spatially heterogeneous, and mean

January–April maximum temperatures alone explain most of the observed phenological variability ($r=0.81$; $p<0.0001$), we feel confident that our approach is suitable and sound. We therefore added the following sentences at the end of the results section “While interannual and longer-term changes in the UK’s FFDs are highly sensitive to January–April temperatures, neither the sufficiently high precipitation totals in winter and spring over the British Isles, nor daylight length play an important role in changes in the onset of the growing season.”.

Result

I think the division between north or south, rural or urban, high or low is too rigid. Why not use latitude, distance from city center, and Altitude as coordinate axis, which will depict more detailed information. The pre-season before PPT may be considered then the fixed month in Figure S6. Except maximum, mean and minimum temperatures, you can try the CV of temperature.

We now better describe the reasons for splitting the FFD dataset into north and south “The 332,107 phenological observations below 53.5° North (around Manchester) were classified as “Southern” (South) and the 87,247 above 53.5° North were considered “Northern” (North) based on differences in the British climate. The northwest is characterised by mild winters and cool summers, the northeast is characterised by cold winters and cool summers, the southwest is characterised by mild winters and warm summers, and the southeast is characterised by cold winters and warm summers. While the west has a more maritime climate during winter, the east is often affected by cold airflow from the European continent. The latitudinal threshold used is informed by floristic regions and the distribution of species in our dataset.”.

We also divided the FFD dataset into 251,315 and 168,039 phenological observations below and above 82 m asl, respectively, which is to the elevational mean of all sites.”. Furthermore, we outline how the dataset was divided into urban and rural subsets “We further differentiated FFD observations into either urban (232,291) or rural (179,897) sites, excluding a total of 7,166 observations from either indistinguishable or of hybrid site type classifications.”. Please note that we re-calculated the elevation threshold and changed all figures and numbers in the text (and figure 3 and table 1) accordingly. In fact, we now use the precise elevational mean of all sites (i.e., 82 m asl) to differentiate between low elevation and high elevation sites. Most importantly, the new splitting criterion did not change our results (figure 3b and table 1).

Discussion

The discussion section does not delve deep enough. The discussion could be improved by emphasizing the final messages.

We added a critical discussion on remaining uncertainties “Possible recording errors and different perceptions of individual observers, spatiotemporal changes in species composition and observer location, as well as ontogenetic effects arising from the long-term monitoring of individual plants cannot be disregarded.”, another reference (32), as well as a concluding statement “If plants in the UK continue to flower earlier, and if the frequency, intensity and duration of climatic extremes increase further, the functioning and productivity of biological, ecological and agricultural systems will be at an unprecedented risk of irreversible socioecological consequences.”.

Referee 2:

The authors investigated the first flowering dates using a large dataset in UK, over a quiet long time period 1753–2019, and they found one month earlier after 1986, and earlier flowering at low altitude and latitudes, as well as in urban area. This is interesting study, and valuable using such impressive dataset. Climate warming advanced spring phenology has been well documented, as well its spatial difference, i.e. along latitude and altitudes. I'm only one question about the methodology, which may damage the results.

We are thankful for this thorough evaluation, have considered all comments and suggestions, and have improved the manuscript accordingly.

The authors found and highlighted that the first flowering dates were 26 days difference between 1735-1986 and 1987-2019, as well as some days difference between high and low altitudes and latitudes, that's interesting, but might be large uncertainty in these estimations, because of larger difference in species composition, phenological sites and records numbers between these two time periods. Although the authors mentioned this limitation in the discussion, but too simple, and need further investigation. To test the robustness, it would be necessary to check these difference at species level, in details, the authors may estimate the diff of FFDs for each species at each site, and then calculate the mean +- std values. In this way, the estimation should be more reliable.

We fully agree that the dataset is noisy, and therefore emphasise the importance of a large sample size in the manuscript (see also previous responses).

Please note our 'biogeographic' data splitting (see table 1 for an overview together with figures S3–S5), as well as the rather conservative temporal restriction to the 1952–2019 period during which both the phenological observations as well as the climatic measurements are most reliable. All possible measures have been performed and implemented to avoid bias in and overinterpretation of our findings. The observed ~26-day earlier flowering in UK plants, however, is a robust finding that is best reflected by using the entire collection of 419,354 first flowering dates (figure 2).

Moreover, figure S3 shows the significant difference of 17 days in those 25 species that have continuous observations of at least 30 years in both old and recent subperiods.

Specific comments

L81-83, how many sites were selected?

We added "We obtained a total of 435,029 spatiotemporally explicit recordings of first flowering dates (FFDs) from 406 plant species between 1753 and 2019 at 6279 'sites' across the UK (see figure 1A for the distribution of all observation points), including the Crown dependencies of Jersey, Guernsey and the Isle of Man, hereafter called UK purely for brevity."

L103-104, why chose 53.5oN as the separate point between southern and northern? same as the 150m as the point between lower and higher elevation?

We now provide further information and re-wrote the section accordingly "The 332,107 phenological observations below 53.5° North (around Manchester) were classified as "Southern" (South) and the 87,247 above 53.5° North were considered "Northern" (North)

based on differences in the British climate: The northwest is characterised by mild winters and cool summers, the northeast is characterised by cold winters and cool summers, the southwest is characterised by mild winters and warm summers, and the southeast is characterised by cold winters and warm summers. While the west has a more maritime climate during winter, the east is often affected by cold airflow from the European continent. The latitudinal threshold used is further informed by floristic regions and the distribution of species in our dataset.”

L112, the resolution of the WorldClim should be provided.

We added “Climate information and site elevation was extracted from WorldClim version 2.1. [18], which is available at 30 seconds spatial resolution (~1 km²).”

L132-134, the figures are replicated with above, here may remove this background information.

We re-wrote the section and removed all methodological description to avoid redundancy.

L210-213, other limitation need to be discussed, for example the ontogenetic difference, as well as the monitored trees, shrubs as well as plant were likely different during different time periods.

We added “Possible recording errors and different perceptions of individual observers, spatiotemporal changes in species composition and observer location, as well as ontogenetic effects arising from the long-term monitoring of individual plants cannot be disregarded.”

L221-222, yes, the direct experimental proof, see Fu et al., 2019 gcb, they reported a photoperiod effect only when the warming treatment after a threshold. Short photoperiod reduces the temperature sensitivity of leaf-out in saplings of *Fagus sylvatica* but not in horse chestnut. *Global change biology*, 2019,

*We added this reference to the main text “[32]”, as well as to the reference list “Fu YH, Piao S, Zhou X, Geng X, Hao F, Vitasse Y, Janssens, IA. 2019 Short photoperiod reduces the temperature sensitivity of leaf-out in saplings of *Fagus sylvatica* but not in horse chestnut. *Glob. Change Biol.* 25, 1696–1703.”*

L246, 5.4 days per decade, provide the time period, I guess it should be 1952-2019, or even late?

We re-wrote two relevant sentences “Based on valuable citizen science and exceeding previous findings at an alarming rate, our analysis reveals an average phenological advancement of 5.4 days per decade between 1952 and 2019. The observed range of annual mean FFDs in this period is more than two months (7th June 1968 versus 2nd April 2019).”

L258-259, the argument is unclear. a community wide shifts in flowering time will be harmful for biology, ecology and society, more details need to be provided.

We re-wrote “If plants in the UK continue to flower earlier, and if the frequency, intensity and duration of climatic extremes increase further, the functioning and productivity of biological, ecological and agricultural systems will be at an unprecedented risk of irreversible socioecological consequences.”

Appendix B

Prof Locke Rowe
Editor *Proceedings of the Royal Society B*
Earth Sciences Centre, University of Toronto, Canada

29th December 2021

Dear Prof Rowe

Please find below our detailed point-by-point responses, and do not hesitate to contact me in case anything remains unclear or further information is needed.

Yours sincerely
Ulf Büntgen

Editor: The main concern here is the heterogeneity in the before and after data sets. I think you have taken some steps in the discussion to emphasize this suggests some caution in interpreting your findings. I would just like to see a little more emphasis on this in the discussion. On the upside, I think your point about S3 could be a good one and should be included in the discussion. I am assuming that "continuously" means each year and in the same sites?.

To further emphasize the heterogeneity of the UK-wide first flowering dataset, we combined figures S3-S5, moved the new hybrid figure to the main text (figure 4), and made all necessary changes throughout the manuscript and its electronic supplementary material. Moreover, we added critical description to the discussion and revised several text parts towards clarification (see marked version of the revised manuscript).

Referee 1: I think authors have done a good work in the revision and all my previous comments and concerns were well addressed. It is now to accept the manuscript.
Many thanks.

Referee 2: Although the manuscript was improved, but the major concern was not solved. The robustness needs to be tested as I mentioned, because of different species composition, different sites and different regions were used between these two time periods, see my previous review. I would suggest the authors to do further test, for example as I suggested in my review, calculate the mean +-std values at species level, and then compare its different

between different time periods. You may also check the difference at species level or using same species between these two periods.

This is now shown in the new figure 4 and related text sections of the revised manuscript.

By the way, the authors only provided the reason why chose 53.50N as the separate point between southern and northern? Why using the 150m as the point between lower and higher elevation? it's not clear. please reply to each comment.

Please note that we are no longer using 150 m asl as the elevational threshold between the lower and higher subsets, and already changed this in the last round of revision during which we added the following sentence "We also divided the FFD dataset into 251,315 and 168,039 phenological observations from below and above 82 m asl, respectively, which is the elevational mean of the total dataset.". Moreover, we adapted all relevant text sections and figure 3c.

Furthermore, I agree with the reviewer#1, a partial correlation analysis, rather simple Pearson correlation is more suitable, and need to be tested as well.

Again, we did compare the FFDs against a range of climatic variables that change from year to year and wrote "Monthly-resolved, 0.25° x 0.25° gridded minimum, mean and maximum temperatures, precipitation totals, and sea level pressure, averaged over the British Isles between 50–60°N and 8°W and 2°E, from E-OBS v23.1 [23], were used for comparison against the phenological data from 1952–2019, the period across which the FFD data are the most robust in terms of mean and variance.". However, since photoperiod does not change at interannual timescales, spring precipitation totals across the UK are both sufficient for growth and spatially heterogeneous, and mean January–April maximum temperatures alone explain most of the observed phenological variability ($r = 0.81$; $p < 0.0001$), we feel confident that our approach is suitable and sound. We therefore added the following sentences at the end of the results section "While interannual and longer-term changes in the UK's FFDs are highly sensitive to January–April temperatures, neither the sufficiently high precipitation totals in winter and spring over the British Isles, nor daylight length play an important role in changes in the onset of the growing season."